# Enabling Better Nutrition for Adolescents from Middle Eastern Backgrounds: Semi-Structured Interviews with Parents

**DOI:** 10.3390/nu13113918

**Published:** 2021-11-01

**Authors:** Nematullah Hayba, Yumeng Shi, Margaret Allman-Farinelli

**Affiliations:** Discipline of Nutrition and Dietetics, Charles Perkins Centre, University of Sydney, Sydney 2006, Australia; yshi7693@uni.sydney.edu.au (Y.S.); margaret.allman-farinelli@sydney.edu.au (M.A.-F.)

**Keywords:** adolescents, parents, eating habits, nutrition, interviews, ethnic minority, overweight, obesity, lifestyle, culturally responsive

## Abstract

The unyielding obesity epidemic in adolescents from Middle Eastern (ME) backgrounds warrants culturally-responsive and co-designed prevention measures. This study aimed to capture the opinions of ME parents residing in Australia on the crisis and their enablers and barriers to healthy eating interventions given their influence on adolescent eating behaviors. Twenty-six semi-structured interviews were conducted with ME mothers, aged 35–59 years, and most residing in low socioeconomic areas (*n* = 19). A reflexive thematic analysis using the Capability, Opportunity, Motivation-Behaviour model and Theoretical Domain Framework was conducted. Parents expressed confidence in knowledge of importance of healthy eating, but were reluctant to believe behaviours were engaged in outside of parental influence. Time management skills are needed to support working mothers and to minimize reliance on nearby fast-food outlets, which was heightened during COVID-19 with home-delivery. Time constraints also meant breakfast skipping was common. A culture of feeding in light of diet acculturation and intergenerational trauma in this diaspora was also acknowledged. Parents pleaded for upstream policy changes across government and school bodies to support parental efforts in the form of increased regulation of fast-food and subsidization of healthy products. Opportunities for weight-inclusive programs including parenting workshops underpinned by culturally-responsive pedagogy were recommended.

## 1. Introduction

Adolescent obesity is forecast to increase worldwide, with 254 million children aged 5–19 years predicted to be living with obesity in 2030 [1]. In Australia, the most recent comprehensive survey from its most populous state, the 2015 NSW School Physical Activity and Nutrition survey (SPANS), revealed that 27.4% of adolescents in secondary school aged 13 to 16 years were either overweight or obese [2]. Moreover, the distribution of obesity is unequal across the socio-economic spectrum, and follows an inverse gradient, with obesity concentrated in populations from low socio-economic status (SES) and ethnic minority backgrounds. In particular, SPANS data indicated that adolescents from Middle Eastern (ME) backgrounds (41%) have a much greater prevalence of overweight and obesity compared with children from English-speaking backgrounds (26%), especially among boys (45% to 27% respectively). A recent secondary analysis of Australian Health Surveys comparing prevalence and period trends in 4 to 16 year olds from 1997 to 2015 demonstrated that, over time, children from ME backgrounds were consistently more likely to be overweight or obese compared with children from English speaking backgrounds [3]. This is concerning, with recent reports indicating 90% of adolescents with obesity will track into adulthood [4]. This places an enormous financial demand on our public health system [5,6], and blights the future of our adolescents with early onset of associated psychosocial and physical co-morbidities [7,8,9].

Despite its concerning trajectory, adolescent obesity is amenable to modification of lifestyle factors such as physical activity, sleep, and diet, with the adolescent period providing a key window for the adoption of healthy behaviors. This is especially relevant for adolescents from ME backgrounds, who were less likely to meet targets in key areas as compared with those from English speaking backgrounds. Risk behaviours included more frequent eating in front of the television (37% vs. 19%), more unrestricted snacking (73% vs. 50%), and a lower likelihood of meeting the recommended daily intake of vegetables (3% vs. 11%). Such a disproportionate distribution of obesity, associated lifestyle risk behaviours, and inadequacy of interventions targeting adolescents, from ethnic minority backgrounds [10,11], illuminates the need for evidence-based, culturally relevant, and locally designed lifestyle programs.

In order to design such a program, ME adolescents themselves should be co-designers in the process, and we have previously published our findings from focus groups that captured their opinions [12]. However, there is also a need to seek the input of other key stakeholders such as parents’ perspectives on weight-related practices of adolescents, particularly from ethnic minority backgrounds. Despite the growing autonomy and independence of adolescents, parents remain an influential stakeholder in shaping the home environment. Thus, it is essential to understand their knowledge, beliefs and attitudes, and impact on uptake of healthy eating behaviours in adolescents from an ME background.

Hence, the primary objective of this study was to explore parents’ perspectives on the childhood obesity epidemic and practices concerning key food behaviors identified as placing ME adolescents aged 13 to 18 years at a greater risk of overweight and obesity. Additionally, their opinions on the efforts needed to tackle it and the design of future intervention programs were explored.

## 2. Materials and Methods

### 2.1. Study Design

This is an explorative qualitative study with parents of adolescents from an ME backgrounds using semi-structured interviews to capture information on weight-related food behaviors, opinions on the obesity epidemic, efforts needed to overcome obesity, and opinions on lifestyle programs and recommendations for future interventions. This qualitative study design was selected to explore parents’ in-depth opinions, and has documented high reliability and validity [13,14]. The findings are reported according to the Consolidated Criteria for Reporting Qualitative Research (COREQ) [15]. This 32-item checklist was developed to promote comprehensive and transparent reporting of interviews and remains the only reporting guidance for qualitative research to have received wide endorsement. This study was approved by the University of Sydney Human Research Ethics Committee (2020/708).

Two inter-related behavioural frameworks were used to develop the scripts for the interviews and subsequent coding and analysis. The Capability, Opportunity, Motivation-Behaviour (COM-B) model [16] posits the performance of a behavior to be the result of an interaction between three drivers—capability (psychological, physical), opportunity (social, physical), and motivation (reflective, automatic). The use of this model allows researchers to identify components that need to be targeted to elicit the behavior, essential for the design of future interventions [17]. The complementary Theoretical Domains Framework (TDF) [18,19,20] offers 14 domains that readily map onto the components of the COM-B model, granting greater insight and depth to this piece of qualitative research. Hence, initial analysis of the interviews via COM-B was further elaborated using the TDF.

### 2.2. Participants and Recruitment

Individuals that met the following criteria: (1) parent to a teenager aged 13–18 years; (2) of ME background; (3) residing in Australia; (4) English-speaking; and (5) able to provide informed consent, were recruited to participate in the study. Recruitment flyers were shared on social media platforms such as Facebook, Instagram, and WhatsApp, which made it available to existing connections, friends, and public groups within the networks of the researchers. Printed recruitment flyers were posted on public noticeboards in the local government areas where parents of ME backgrounds congregate, e.g., local community hubs and recreation centers. In addition, passive snowball techniques were used, whereby eligible participants who had joined the study informed others about the study.

Participants scanned a QR code located on the flyer that directed them to a screening questionnaire on the Research Electronic Data Capture (REDCap, Vanderbilt University, Nashville, TN, USA, 2004). If eligible, they continued to complete a short demographic questionnaire that collected how many children aged 13–18 years they have, the age of their children, their age, postcode (indicative of socio-economic status), email, gender, and whether or not they are a single parent. Postcode was used to assign the Socio-Economic Index for Disadvantage and Advantage Area (SEIFA), as determined by the Australian Bureau of Statistics (ABS) [21]. Upon completion, participants were sent an automatic email with the participation information statement and consent form. All participants provided informed consent. One researcher (N.H.) organised dates and times for the semi-structured interview conducted via telephone or Zoom, as selected by the participant.

### 2.3. Procedure and Data Collection

The lead researcher (N.H.), who is a female accredited practicing dietitian and PhD Candidate, and of Middle Eastern background, conducted the semi-structured interviews, which ran for about 60 min. The semi-structured interview on perspectives on food behaviors and opinions on lifestyle interventions used a guide of 100 questions and 7 probes, developed a priori (Table 1). The questions were arrived at by the consensus of two experienced researchers (N.H. and M.A.F.) and structured on the COM-B [16] model and the TDF [18,19,20]. Probes were only asked if a participant’s response to the main question needed further exploration.

At the start of each interview, a few minutes were allocated for a brief introduction and building rapport. Participants were also informed that their personal data would be kept confidential and asked for final consent for recording of the interview. Following this, the researcher began the session with two open-ended questions asking participants to reflect on their parenting experience as a whole in relation to their adolescent’s health. They then provided parents with the current statistics on the obesity pandemic and asked for their opinions on its existence and the efforts needed. It then proceeded to questions on perceptions and practices on four food behaviors—(1) eating breakfast; (2) eating fruits and vegetables; (3) soft drink (carbonated sweetened beverages) consumption; and (4) fast food consumption. The study topics were determined by the key nutrition risk behaviours, as identified by the SPANs survey [2]. After completion of the interview, each participant was sent a $20 (AUD) electronic voucher for their time.

### 2.4. Data Analysis

Interviews were digitally audio-recorded and a two-step transcription process was used, consisting of the initial main transcription verbatim and a second hearing to correct any errors. The transcripts were not returned to study participants. Transcriptions and audio data files were stored on the Research Data Store (RDS, University of Sydney, Sydney, Australia, 2020/21). NVivo Pro (12th Edition, QSR International, Doncaster, Australia, 2018), a qualitative data analysis software, was used for analysis of the de-identified transcripts. A reflexive thematic analysis was performed to construct the coding with both deductive (themes arising from the interview questions) and inductive approaches (themes emerging from the data during analysis). Two researchers conducted the coding following a five-step process consistent with framework analysis [22], as follows: (1) the two researchers independently analysed the transcripts (N.H. and Y.S.); (2) preliminary coding was developed by each researcher using the themes from the COM-B model and the TDF and any new themes that emerged; (3) each researcher assigned a code to all data; (4) the researchers met and then arrived at a consensus for the coding, and then charted all the data; and (5) the data were mapped and interpreted and representative quotes were selected for each theme.

## 3. Results

Of the 58 people that completed the screener questionnaire, 57 were eligible. Of the 57, 26 took part in the interview process. The recruitment period was from November 2020 until July 2021. Recruitment of participants continued until the lead researcher conducting the interviews determined that no new topics were emerging and an information-rich data set was obtained. The interview duration ranged from 40 min to 1 h 33 min. All interviews were in English, with one exception of a parent more comfortable speaking in a combination of English and Arabic. The majority of parents were aged 40–49 years (n = 17), parent to one or two adolescents (n = 20), aged 13–14 years (n = 14), not a single parent (n = 21), and from low to middle SES areas (n = 19), as determined by SEIFA. Demographic information of the sample is displayed in Table 2.

The findings are summarised below under the six major themes of the COM-B model and subthemes from the TDF, with relevant quotes presented in Table 3. No part of the interview was coded to memory, attention and decision processes, intentions, beliefs about capabilities, and automatic motivation for social/professional role and identity.

### 3.1. Psychological Capability

#### 3.1.1. Knowledge

Obesity pandemic: Most parents agreed there was a current obesity crisis in adolescents from an ME background. Parents displayed a high level of awareness and knowledge of the need for healthy behaviours and were cognizant of the extent to which their adolescents adopted them. Furthermore, some indicated the need for government guidelines for healthy eating to be translated into actionable and culturally relevant resources. Some also called for parenting workshops, considerate of feeding practices in light of intergenerational trauma. A minority of parents expressed a dissident view, being that ME adolescents were more affected by obesity and believing children were physically active and unaware, stating that most of the adolescents they knew were quite active and of healthy weight. If there was an obesity crisis, it was not exacerbated in the ME community.

Breakfast skipping: Breakfast was considered extremely important in the ME culture; however, a traditional breakfast remained elusive given time constraints. Some parents were convinced that, because of their experiential learning such as fasting in Ramadan or intermittent fasting, delay of breakfast was relatively safe and healthy. Parents claimed that their children knew of the importance of breakfast; however, skipping breakfast was still common. Several consumed Manoush—a very popular Levantine flatbread purchased on the way to school.

Fruits and vegetables: All parents indicated the importance of fruit and vegetable intake, but most had limited knowledge on how to counteract their adolescent’s fussy eating, especially their reluctance to consuming vegetables. Some expressed a desire to learn more healthy ways of snacking to fulfil children’s hunger. The need for a variety of vegetables was acknowledge and parents tried their best to incorporate them into meals, as well as including a salad when vegetables were not inherently present in the main dish.

Sugar-sweetened beverages (SSBs): Parents are highly sentient of the ‘dangers’ of SSBs and believe avoidance or any reduction will be beneficial for their adolescents. Parents who did not prohibit soft drinks reported that their children consumed soft drinks at least three times weekly. Others replaced sugar-sweetened soft drinks with sugar-free soft drinks, and allowed daily consumption. Parents were unsure of the exact benefits or possible long-term effects of sugar-free beverages except that they were sugar-free. The occasional fruit juice beverage was also used as a substitute beverage. One parent advised of the school principal’s advocacy by holding a knowledge workshop to educate parents on the dangers of energy drinks, which enabled her to ban it from her children. Prior to this workshop, her knowledge on SSBs was restricted to soft drinks.

Fast food: Parents understood the long-term consequences of continous fast food consumption, but also acknowledged its appeal as a quick, cheap, and easier option for the majority of working mothers. Hence, fast food was considered a guilty pleasure by most at least once a week where the mother did not have to cook. Some believed fast food could be included in moderation, ranging from once a month to three times per year. Participants distinguished different forms of eating out and fast food, claiming some to be inherently healthy and not pose a risk when contrasted to others such as ‘Mcdonalds, KFC, and Oportos’. Healthier options included charcoal chicken from local favourites such as ‘Awafi chicken’ and wraps and salad from ‘Lewrap’, ‘Douggies’, and ‘Subway’.

#### 3.1.2. Behavioural Regulation

Breakfast skipping: Parents highly encouraged their children to eat breakfast to ensure a healthy start to their day. Most parents sought to have their children have a healthy meal, but were satisfied if the reluctant adolescents had at least one piece of fruit or a breakfast replacement beverage such as ‘Up and Go’. Parents reported that they did not force adolescents, but tried their best to encourage this meal.

Fruits and vegetables: Parents encouraged their children to consume vegetables by constantly including them in their meals, packing them into lunchboxes, and preparing cut up vegetables as snacks and making them available in the home. Parents also included vegetables by making salad sandwiches and adding them into pasta sauces. Parents that promoted vegetable consumption when adolescents were younger found it easier to ensure adolescents implemented this practice.

SSBs: Parents regulated soft drink behaviours by controlling the amount brought into the home. If parents saw children bring home soft drink they would kindly remind them of their limits and possibly substitute it with juice. Many simply banned and replaced them with non-sugar alternatives such as cordials and flavoured waters and only bought the sugar version for occasions such as barbeques, visitors, and Eid celebrations. Lastly, if parents allowed soft drinks, they regelated their use, not allowing them, for example, to be taken to school and consumed in the morning.

Fast food: For younger adolescents, parents limited fast food consumption by choosing not to purchase and not providing them with money to purchase food products. They did include it occassionally to encourage moderation in lifestyle behaviours and as a relief from continous cooking. They also attempted to recreate healthier versions of fast food at home such as healthy burgers and fried chicken.

#### 3.1.3. Skills

Parents displayed skills in innovative methods in incorporating fruits and vegetables into meals. Parents also displayed great skills in identifying alternative fast food choices and introducing substitute drinks and the promotion of water. Despite this, the interviews revealed a severe deficit in time management skills, given the time-poor nature of these households. Parents communicated the need for help in sustaining healthy, easy meals and tranlsating generic health advice into culturally relevant and actionable recommendations.

### 3.2. Physical Capability

#### Skills

Parents were able to cook healthy and nutritious meals and only relied on fast food when they wanted a break. The most common factor affecting the ability to cook was lethargy from a hectic and busy lifestyle and some illness. Most parents prepared breakfast, but indicated that older adolescents had the skills to assume responsibility for this. Parents mainly prepared fruits and vegetables.

### 3.3. Reflective Motivation

#### 3.3.1. Beliefs about Consequences

Breakast skipping: Parents had very similar beliefs regarding the health consequences of skipping breakfast. They explained the need for a nutritious meal in the beginning of the day for energy and vitality. They believed that, if breakfast was skipped, this would lead to low focus and troubles concentrating in school, exacerbated hunger in the afternoon, and increased bingeing and consumption of energy-dense, nutrient-poor (EDNP) foods upon returning home and at dinner time. This could lead to weight gain.

Fruits and vegetables: While parents believed in health consequences for not consuming enough fruits and vegetables, the outcomes were varied, with some reporting increased weight, lack of vitamins and minerals, weak bones, being displaced for EDNP foods, bloating, impacted performance, pale skin, and consitpation due to lack of fibre.

SSBs: Perceived consequences of excessive SSB consumption were more pronounced, with parents referring to them as ‘poison’, and not considering them a drink. Health consequences such as excessive weight gain, tooth decay, stomach ulcers due to acid, and health complications such as diabetes and weak bones were reported. Mood changes were also identified. In general, parents were very conscious of the repercussions, irrespective of the amount of consumption.

Fast food: Similarily, parents held strong views on the detriments of excess fast food consumption, with some calling it ‘disgusting’ and ‘rubbish’. They reported excess weight and displacement of healthy core foods such as fruits and vegetables as possible consequences. Others also expressed concerns over cleanliness, true food composition, food allergies and asthma, and the presence of food additives and preservatives such as monosodium glutamate as causes of health complications.

#### 3.3.2. Social/Professional Role and Identity

Parents did not shirk, but rather embraced the integral role of parenthood in shaping and informing adolescent dietary behaviours. However, they heavily communicated the need for government bodies and schools to have a more defined role in supporting parental efforts by minimising access to fast food and SSBs, subsidizing healthy products, and providing further opportunities for education through school and dietary lifestyle programs. They also highlighted stricter regulations on fast food advertising and called for more efforts to present healthy food as tasty and enjoyable via government health campaigns. Education and parental influence was deemed insufficient in fostering healthy habits, and needs to be augmented with policies to control and minimise harms of the obesogenic environment that adolescents find themselves navigating as they reach heightened independence, especially in the later age bracket of adolescence (15–18 years of age). If this is not provided, parental guidance was seen as futile in the face of societal and environmental pressure.

All parents believed that they had an important role in promoting breakfast, fruit, and vegetable consumption and moderating SSB and fast food intake. However, some believed that their role dissipitated when adolescents reached 15 years and over, claiming that adolescents were now self-sufficient and needed to take responsibility for upholding healthy behaviours.

#### 3.3.3. Goals

All parents identified healthy eating behaviours of promoting breakfast, fruit, and vegetable consumption and moderating SSB and fast food intake to be of high priority. However, some believed that older adolescents needed to take ownership of these goals and felt less responsibility as a parent for this reason.

#### 3.3.4. Optimism

All parents asserted confidence that making necessary changes such as reducing fast food intake and increasing fruit and vegetable intake will bring about benefits for their adolescent’s health. Similarily, most parents were highly confident in their adolescent’s knowledge of the importance of healthy eating; however, they were less optimistic that adolescents would practice these behaviours in environments outside of parental influence. This was seen in particular for the consumption of soft drink and energy drinks, which was mostly banned in the household, but susceptible to the influence of friends and gatherings outside of the home.

### 3.4. Automatic Motivation

#### 3.4.1. Emotion

Obesity pandemic: Parents communicated their deep fears in attempting to raise healthy teenagers in an obesogenic environment. Irrespective of weight concerns, parents did express worry about the implementation of healthy eating behaviours outside of the home. They emphasised that they wished to impart life-long healthy choices that will be autonomously upheld and of their children’s own volition. When adolescents did not partake in healthy behaviours, parents felt extreme guilt. This is especially true for adolescents that have gained independence and no longer rely solely on parental provision of food.

Breakfast skipping: Parents expressed that their children were primarily not interested in breakfast. Consequently, many parents were concerned regarding their children’s energy and focus levels and delayed hunger, resulting in the consumption of EDNP foods from school or nearby convenience stores and fast food outlets. Parents indicated that they recevied minimal appreciation from children for preparing breakfast as it was seen to be routine.

Fruits and vegetables: Parents expressed that, while their children loved enjoying a variety of fruits, they showed aversion to vegetables, particularly boys. Boys were collectively characterised as fussy eaters, disliking the taste and texture of vegetables, with some parents stating their immediate preference for fast food. One parent indicated that, while fruit can be freshly eaten, vegetables need to be prepped and cut, which her adolescents were too ‘lazy’ to do. Most parents claimed that, upon preparing and making available fruits and vegetables in the form of platters, their children would consume them and show appreciation. Minimal conflicts occurred relative to other food behaviours, and they mainly occurred at the dinner table, with parents trying to encourage their children to eat the vegetable portion of the meal.

SSBs: Interviews revealed that most parents believed that their adolescents were ‘addicted’ to consuming sugary beverages and, accordingly, held extreme dislike for soft drinks owing to awareness of their health consequences. Some reported conflicts did occur with adolescents wanting to consume sugary beverages or upon parents’ discovery of the adolescent’s consumption of sugary beverages outside of the home.

Fast food: Similar to soft drinks, parents were disturbed by the prevalence of fast food intake and their adolescent’s easy and prevalent access to it as a result of its close proximity to residential neighbourhoods and remote delivery systems such as Uber. Parents expressed concern for their food choices, with adolescents divorcing themselves from their traditional cultural food served at home and preferring burgers and pizza to satisfy their cravings.

#### 3.4.2. Optimism

Parents voiced strong confidence in knowledge of the benefits of lifestyle changes to adolescent health. Correspondingly, most were optimistic that adolescents were aware of the importance of said behaviours, but less so that these healthy practices will be implemented in the long run and outside of parental influence.

#### 3.4.3. Reinforcement

For eating behaviours, parents rarely applied punishments and rewards. Rather, they applied strict rules such as only sugar-free drinks allowed in the home, and fast food only allowed once per week. If children broke the rules, the parent discussed with them the consequences and ensured that no further products were consumed. Parents reinforced breakfast and fruit and vegetable consumption by preparing meals and encouraging their children to consume them, especially for younger adolescents aged 13–15 years.

### 3.5. Social Opportunity

#### Social Influences

Obesity pandemic: The majority of parents reflected on their parenting practices by drawing attention to the culture of “feeding”, whereby their parents (first generation immigrants) in turn associated overfeeding as an act of love and care, common in ethnically diverse communities. Some parents reflected further and posited this culture to stem from a psychological need for their parents to compensate for intergenerational trauma. With many of their parents fleeing poverty, war, and violence, plentiful food and feeding is linked to generosity, care, and abundance. It was also recognised that immigration to Australia and the resulting diet acculturation meant that second-generation parents shifted from the initially healthy Mediterranean diet of their ancestors and incorporated other foods and cuisines that are not as healthy. Parents identified that influence from friends and social media did contribute to body image, in both males and females.

Breakfast skipping: Most parents reported nil influence from friends on adolescent’s eating breakfast. One did report not eating breakfast because of attempts to lose weight, citing a possible influence from friends. Religious influence was also reported, given fasting practices in Ramadan, which allowed some parents to think that skipping breakfast occasionally was okay.

Fruits and vegetables: Parents reported limited influence of friends and family on the intake of fruit and vegetables. Some parents did report that their daughters were embarrassed to take pre-cut vegetables and fruits to school.

SSBs: Friends and ocassions such as family gatherings and barbecques encouraged the consumption of soft drinks. Energy drinks such as V, Mother, and Redbull were identified by parents to be part of a drinking culture present in schools, more likely in boys. Parents also identified that, when friends spent time with each other, they most likely ate fast food and linked this to soft drink intake, given that many fast food meals and combos sport a soft drink as part of the package.

Fast food: Fast food consumption in adolescents was attributed to opportunities fostered via a combination of social media exposure and interest from friends. Parents reported that the widespread use of platforms such as Instagram and Snaphchat meant that their children were much more aware of new franchises opening up and more suscpetible to peer pressure and influence, with adolescents wanting to try different foods in fear of missing out.

### 3.6. Physical Opportunity

#### Environmental Context and Resources

Obesity pandemic: Parents reported interventons such as Go4Fun [23] and Crunch and Sip [24] for their children in primary school, but opportunities for adolescents were restricted to cooking and Personal Developement, Health, and Physical Education (PDHPE) classes. Parents emphasised the importance of such school and government programs to continue into high school and be further embedded into the currciculum. One parent highlighted the provision of an adolescent obesity weight loss program administered at Westmead hospital, which was instrumental in allowing both of her adolescents to reach a healthier weight, and called for greater awareness of these programs. Some identified local workshops run by dietitians to be of great value and encouraged more of these opportunities.

Breakfast: Parents identified that the main physical barrier was lack of time owing to poor time schedules. With the majority of parents being working mothers coming from low SES areas, parents identified the rush in the morning to get to school to hinder breakfast eating practices. Breakfast (small or traditional Arab) was more likely to be provided and eaten during weekends, where more time was available. Many reported the presence of breakfast clubs at schools and encouraged adolescents to eat if they missed breakfast at home.

Fruit and vegetables: Parents reported lack of time and limited financial resources, mostly claiming that healthy food was generally expensive as opposed to fast food, which was cheap and more time feasible, not requiring preparation and cooking time.

SSBs: Most parents influenced the home environment by not purchasing sugary drinks and instead opting for sugar-free versions, cordials, and flavoured sparkling water. While schools have recently removed sugar drinks owing to the NSW government’s Healthy Canteen Strategy [25], convenience shops on the way to school were reported to enable easier access to SSBs.

Fast food: Parents reported that, when adolescents gained physical and financial independence, fast food intake increased dramatically, especially in boys. Financial and time restraints made fast food a more feasible option, despite awareness of health consequences for working mothers that did not have time to cook regularly. Fast food consumption in adolescents was also linked to the close proximity of fast food outlets to areas of residence, particularly those from low SES neighbourhoods. Parents also identified remote food delivery services, i.e., UberEats, to be a powerful enabler, given its access to adolescents even during school time. This was noted to be exacerbated during the COVID-19 pandemic.

## 4. Discussion

Despite the greater risk for overweight and obesity among adolescents from ethnic minorities, co-designed prevention research to effectively intervene and halt the early emergence of chronic disease in this priority population remains scarce. The current findings suggest that the parents’ knowledge and skills about what constitutes healthy eating and links with obesity are overall good. However, they grapple with a food environment and peer pressure that exert strong, but negative influences on their adolescents’ nutrition behaviours. Parents themselves said other barriers were their own lack of time for preparing home cooked meals and some cultural and generational traumatic experiences influenced them to use food as a sign of nurturing.

While the underlying mechanisms for the predisposition of migrants to an increased risk of obesity and its related comorbidities are complex and multifaceted [26], the literature suggests that lifestyle changes and dietary acculturation are key factors [27]. A recent secondary analysis of Australian national data revealed multi-generational inequalities in overweight and obesity linked to greater acculturation, with males arriving as adolescents from North Africa/ME and Oceania regions having significantly higher BMI and being twice as likely to be overweight and obese compared with immigrants that arrived as adults [28]. A systematic review assessing acculturation and weight gain among Latino youth in the United States provided preliminary evidence that, unlike preschooler and elementary age groups, second-generation adolescents were more likely to be obese compared with first-generation adults, indicating a positive association between U.S. acculturation and obesity [29]. A greater degree of acculturation is also negatively associated with fruit and vegetable, meat, and grain consumption [30], and is positively associated with SSBs, saturated fat, and energy from discretionary foods [31]. In addition, parents utilise feeding practices that have evolved over thousands of years, encouraging children to eat despite not being hungry, in response to distress, often using force feeding and coercion to promote necessary intake to support growth. However, in conjunction with food insecurity and the current obesogenic environment, these traditional feeding practices promote weight gain and overeating [32].

The majority of the participants in this study came from a low SES area and complained of a highly obesogenic environment with plentiful nearby fast food outlets. It has been demonstrated that fast food outlets are more frequently found in lower SES areas in Australia [33]. Online food delivery (ODF) such as UberEats^®^ intensified the availability of fast food intake, and recent evidence suggests that 80.5% of menu items are discretionary food items [34]. Selection of fast food is exacerbated in the context of diet acculturation, as migrant children move away from the traditional cuisine and integrate the Western diet and grow to love fast food. It has been shown that there is a positive correlation between obesity and acculturation, with adolescents who have assimilated or been highly integrated into the dominant culture being more likely to adopt food habits such as increased fast food, SSBs, and other EDNP foods [35,36,37,38], which are linked to higher rates of obesity and related co-morbidities. Furthermore, a recent comprehensive secondary analysis of the National Health and Nutrition Examination Survey (NHANES) has suggested that, as they become more acculturated, Asian American youth are more likely to frequently consume fast food [39]. Moreover, gender differences were noted, with acculturated Asian American boys, unlike girls, more likely to be obese. Parents in our study did convey their distress regarding adolescent consumption of fast food and SSBs, and this was supported by the SPANs data [2]. Similarly, a recent qualitative systematic review investigating cultural influences on childhood obesity and parental practices in ethnic minorities identified factors that resonated with our study [40]. Child-feeding practices, whereby parents encouraged children to eat until they were full or their plates were finished, as well as indulgent feeding (catering to child’s tastes and wishes), were identified as a consistent theme across studies, which also emerged from our analysis [40]. These practices have been linked to higher BMI [41,42].

Conversely, the review also identified family meals to be a constant feature, which has been shown to be protective of obesity regardless of ethnicity, but also because of the opportunity it presents to prepare and consume traditional and ethnic foods [40]. In a study of African migrant children in Australia, it was demonstrated that children that maintained traditional eating practices had a significantly lower BMI than other cultural affiliations such as marginalised (neither host or origin culture) and integrated (both origin and host) children. They also consumed a less energy-dense diet than their marginalised counterparts [43].

Parents in this study indicated a readiness to adopt programs for healthy lifestyles and indicated that they saw this as a shared responsibility of governments and schools. They were aware of state-run programs already on offer to younger children. In contrast a previous study in Victoria, Australia reported that less than half of the parents from disadvantaged communities were ready to embrace community obesity prevention programs, and they had a poor level of knowledge concerning childhood obesity [44]. Another Victoria-based study reported on semi-structured interviews with migrant families from ME, Vietnamese, Indian, and African countries, and indicated knowledge about obesity prevention was poor and that there was a general reluctance to engage in programs [45]. Thus, these ME parents in Sydney seemed more informed, but what is apparent from all three studies is the need to engage with the relevant stakeholders when one is attempting to create a program that meets the needs of ME adolescents and their families.

The importance of supportive food environments must also be emphasized. Zoning regulations that prevent the proliferation of fast food outlets and advertising close to schools in socioeconomically disadvantaged areas would enable parents to exert more influence over their adolescents’ food intakes outside the home. Social media environments were also viewed as exerting negative influences on adherence to a healthy diet as peers swapped images of new restaurants and adolescents wanted to participate in trying the food outlets. A recent scoping review of adolescent behaviours in eight countries, including Australia, highlighted the influence of social media and the desire for peers to conform, but also found it could be used to promote healthy eating [46].

Common issues were identified between parent attitudes and practices identified in this research and adolescent perspectives explored in focus groups in a sister study [12]. Adolescents who ate more meals prepared at home were more likely to consume healthy meals. This corresponds with the behavioural regulation of parents who tried their best to prepare healthy meals including fruits and vegetables to provide healthier diets for their adolescents. Both adolescents and parents reported fast food consumption typically occurred on a weekend when parents needed a break. Adolescents also reported the availability of snacks to trigger cravings for consumption and parents reported the need to limit items such as sugary drinks in the home to mitigate these cravings and limit accessibility. Lastly, both adolescents and parents highlighted the role of social influence in increasing opportunities to consume less healthy fast food and soft drinks. These commonalities underscore the importance of interventions that enable ownership by adolescents, but also involve parents allowing parent–adolescent interactions to achieve healthy dietary behaviours.

### 4.1. Implications for Practice, Policy, and Research

The failure of interventions to generate any effect on the incidence of obesity in adolescents [11], especially those from ethnic minority backgrounds [10], has warranted the identification of other potential factors influencing obesity rates in this at-risk population. Factors such as SES, diet acculturation, and food insecurity highlight the need for a localised approach to be implemented with policies targeting social disadvantages and interventions rooted in culture-specific traditional dietary habits to be researched and conducted. To ensure culture congruency and safety, co-design is essential to allow at-risk communities to champion lifestyle interventions. Similarly, the dearth of qualitative studies conducted in parents of adolescents, in particular from ethnic minority groups outside of the United States, illustrates the critical need for more studies to be conducted to build the evidence base for this priority population. Of the 12 studies reviewed in the recent qualitative systematic review, all were conducted in the United States, mostly with Latino/Hispanic ethnic groups [40], except for one conducted in Australia, which included Vietnamese, Burmese, Afghani, South Asian, and African ethnic/cultural groups [47] and with parents of children. Only one was conducted with adolescents and their parents and examined physical activity behaviours only [48].

### 4.2. Strengths and Limitations

This study has several strengths, including the rich data set provided by using semi-structured interviews and the careful coding by two researchers with a third to discuss themes. Among the limitations is that the findings can only be generalized to this population, and the fact that these parents were possibly somewhat more advantaged than the migrant ME parents in Melbourne could account for observed differences. All interviews were with mothers and most from families with two adults. This could be because of gender stereotyping, where mothers are traditionally considered the primary caregiver responsible for meal preparation. Fathers are also influential with respect to what their children eat [49].

## 5. Conclusions

The findings support the development of a culturally relevant program that takes into consideration the experiences unique to immigrant populations such as diet acculturation, food insecurity, and traditional feeding practices. Parenting workshops that are rooted in the maintenance of traditional cultural orientation in the context of an obesogenic environment are recommended. Government initiatives are needed to facilitate equity-based policies to neutralize the interplay between low SES and food access to enable at-risk populations to adopt healthy eating behaviors. The findings cannot be generalized to other groups outside Australia, but may serve as a model for qualitative research design in ME populations elsewhere in the world.

## Figures and Tables

**Table 1 nutrients-13-03918-t001:** Questions used in interviews to gather information on the capabilities, opportunities, and motivations of parents on adolescent food behavior and perception of the obesity pandemic.

Study Topic	Questions
Introduction	To start, could you please tell me a bit about your experience as a parent to an adolescent?
	Probe around role of parenthood in health of adolescent
	In general, do you have any concerns about his/her lifestyles?
	Probe for concerns about body weight.
	Probe for opinion on excess weight.
Perception of the Obesity Pandemic	The 2015 NSW School Physical Activity and Nutrition Survey (SPANS) revealed that adolescents from Middle Eastern Backgrounds have a much greater prevalence (41%) of overweight and obesity compared with children from English-speaking backgrounds (26%) especially amongst boys (45 to 27% respectively).
	What do you think about these statistics?
	Probe for agreement/disagreement.
	If yes, why do you think this is the problem?
	Who do you think is responsible?
	Who do you think has a role in regulating this?
	Probe for schools, community groups, health professionals, government, parent themselves.
	What do you think should be done?
	Probe for education, lifestyle programs etc.
	Have you participated in any sort of health intervention for your child?
	Probe for education, lifestyle programs etc.
	Do you have any areas of improvements or suggestions for the intervention?
Target Behaviour 1: Eating Breakfast	What do you think about eating breakfast? Does your child eat breakfast?
	Why/Why not?
	Is making breakfast something you usually do?
	Who usually makes breakfast?
	Does your child know how to cook breakfast?
	Does your child often skip breakfast?
	Do you know if your child has breakfast every day?
	Do you see preparing breakfast for your family as an important part of your role as a parent?
	If you were to rate from 1 to 10 on how important it is, what would you say?
	Do you have certain number of days they should have breakfast?
	What do you think will happen with your child’s nutrition, body weight if they skip breakfast?
	Does it upset you to worry about morning meal times with your child?
	Are there any emotional reactions from your child about eating breakfast?
	How does that affect you?
	Do you receive any emotional reward or satisfaction from your child for organising breakfast?
	How confident are you to ensure that your child knows the importance of eating breakfast?
	To what extent do the resources available affect your ability to prepare breakfast?
	How much do you think their friends influence whether or not they eat breakfast?
	How much do you think their friends influence what they eat for breakfast?
Target Behaviour 2: Eating Fruits and Vegetables	What do you think about eating fruits and vegetables?
	Does your child eat fruit and vegetables?
	Why/Why not?
	Are fruits and vegetables usually included in meals at home?
	Do you know how to prepare fruit and vegetables?
	Do your children know how to prepare fruit and vegetables?
	What do you do to try and encourage your child to eat more fruits and vegetables?
	Does your child avoid having fruit and vegetables?
	Is there any routine at home with eating fruit and vegetables?
	Do you see including fruits and vegetables as part of meals for your family as an important part of your role as a parent?
	If you were to rate from 1 to 10 on how important it is, what would you say?
	Do you have certain number of days they should eat fruits and vegetables?
	What do you think will happen with your child’s nutrition, body weight if they do not have enough fruits and vegetables?
	Does it upset you to worry about fruits and vegetables with your child?
	Are there any emotional reactions from your child about eating fruits and vegetables?
	How does that affect you?
	Do you receive any emotional reward or satisfaction from your child for organising fruits and vegetables?
	How confident are you to ensure that your child knows the importance of eating fruits and vegetables?
	To what extent do the resources available affect your ability to prepare fruits and vegetables?
	How much do you think their friends influence whether or not they eat fruits and vegetables?
	How much do you think their friends influence how much fruits and vegetables they eat?
Target Behaviour 3: Soft Drink Consumption	What do you think about soft drink consumption?
	Does your child drink soft drink?
	How often does your child have soft drink?
	How much soft drink does your child have per day?
	Do you have any rules about how much soft drink your child can drink?
	What strategies have you used to reduce soft drink intake with your child?
	Do you impose any limits on how much soft drink your child can consume?
	Are there any times that you don’t allow soft drink?
	Do you believe you have an important role in moderating your child’s soft drink intake?
	If not, do you think that you should?
	What might be your plan in this area?
	If you were to rate from 1 to 10 on how important/relevant it is, what would you say?
	What do you think will happen to your child’s health if they were to consume excessive soft drink?
	What do you think is a reasonable amount of soft drink per day?
	Does your child’s soft drink intake affect you emotionally?
	Are there any conflicts regarding soft drink?
	How does this affect you and your child?
	Do you enforce any routine around soft drink?
	Do you give any punishments or rewards to your child for following any soft drink rules?
	How confident are you that regulating soft drink will have a benefit for your child?
	Do you believe your child’s environment affects his/her soft drink consumption?
	To what extent do the resources available to you affect your ability to regulate your child’s soft drink consumption?
	How important is soft drink amongst your children and their peers?
	How difficult does it make to regulate your child’s soft drink intake?
Target Behaviour 4: Fast Food Consumption	What do you know about eating fast food?
	Does your child eat fast food?
	How often does your child have fast food?
	How much fast food does your child have per day?
	What do you think your children know about eating fast food?
	Do you have any rules about how much fast food your child can eat?
	If you wanted to reduce fast food intake, how would you go about doing that?
	Do you know how to cook?
	Do you impose any limits on how much fast food your child can consume?
	Are there any times that you don’t allow fast food?
	Do you have any alternatives for fast food?
	If you wanted to cut down on fast food how would you do that?
	Do you believe you have a role in your child’s fast food intake?
	Do you think you should?
	On a scale of 1 to 10, how important is your child’s fast food intake to you?
	What do you believe will happen to your child’s health if they were to consume excessive fast food?
	How much fast food intake do you think is reasonable?
	Does your child’s fast food intake affect you emotionally?
	Does it affect you and your child?
	Do you enforce any routines around fast food intake?
	Do you give any punishments or rewards to your child for following any fast food rules?
	How confident are you that regulating food intake will benefit your child’s health?
	Do you believe your child’s environment affects his/her fast food consumption?
	To what extent do the resources available to you affect your ability to regulate your child’s fast food consumption?
	How important is fast food amongst your children and their peers?
	How difficult does it make to regulate your child’s fast food intake?

**Table 2 nutrients-13-03918-t002:** Demographic information of interview participants *(n* = 26) *.

Demographic Characteristics	Number of Participants (*n)*
Age group	
35–39	6
40–44	8
45–49	9
50–54	2
55–59	1
Female Gender	26
No. of Adolescents	
1	11
2	10
3	3
4 or more	0
Age of Adolescents (years)	
13–14	15
14–15	3
15–16	3
16–17	8
17–18	7
Single Parent	
Yes	5
No	21
SEIFA ^1^ Rank within Australia—Decile [21]	
<5	19
≥5	7

* *n* stands for sample size; ^1^ SEIFA: Socio-Economic Indexes for Areas (measure of socio-economic status) [21].

**Table 3 nutrients-13-03918-t003:** Exemplary quotes from parents by key theme.

**Psychological Capability: Knowledge**
“I think it’s a matter of teaching Middle Eastern people or parents or caretakers that they can’t smother them. They can’t feel sorry for them because they’re crying, because they want a bag of chips or they can’t go and do fitness with them and take him and feel good about aww he’s just worked out I’ll go get him Maccas. I think it’s about changing the parent’s mindset, about what they do and they turn around and Middle Eastern parents will say to you, I don’t you know, he doesn’t have this, but the whole pantry is full of things that child shouldn’t be eating because they’ve got other kids in the house that want all that stuff and they’re not obese, so they’ve probably got one out of three kids that are obese, And how do you change the lifestyle of that child if you’re not changing the whole house?”—P11, (F), 45 y, >1 adolescent “It’s very important you can’t start your day without eating healthy food to start your day and imagine a car you can’t start it without having petrol in it. So it’s the main thing. But I mean, X rarely eats breakfast, but I encourage her always to eat even if she drink a cup of milk and eat some cereal, anything. You start your day not to start it without food or healthy food. It’s very important”—P23, (F), 48 y, 1 adolescent “It’s crap for you. It’s very, very bad for you. But it’s cheap. It’s easy and it’s fast. When you’ve done a 12-h shift, you really don’t want to come home and cook”—P14, (F), 43 y, 1 adolescent “I think they’re absolutely essential I yeah, we make sure that, you know, there’s salad on the table pretty much every day, if there’s not a salad, you know vegetables cooked in another form, but they’re really, really important”.—P7, (F), 42 y, 1 adolescent “And I’m a hundred and ninety, four hundred percent against those drinks. And I said to him, why are you drinking this? Like what in the world? How many times have I told you not to not to drink it and not to have any of these energy drinks? They’re not good for you and you don’t know what your body reacts is going to react to them and he’s like it’s only a V you know. So it’s like, no, it’s not just only a V (branded energy drink) do you understand what goes in there. Like, seriously, you know, your body is like you’re young”—P8, (F), 46 y, 1 adolescent
**Psychological Capability: Behaviour Regulation**
“I know on the weekends they have their breakfast; they make sure they have their breakfast but yeah during the week I make it as smooth as I can for them to have breakfast”—P1, (F), 49 y, 1 adolescent “Well, a lot of times that I know they’re eating vegetables because I blend it in the nutribullet. Yeah. And if I’m making like, for example, spaghetti is the best because you got the meat. So anything that has, like, little meat mince in it I actually put vegetables through it so they’ll be eating eggplant and zucchinis and everything, Mushrooms”—P19, (F), 45 y, 1 adolescent “Well, obviously you wouldn’t give them money to purchase it. You’re in control. They’re not in control. So if you want them to get McDonald’s then you’re driving, you’re paying. So ultimately you’re allowing me to do or eat unhealthy unless they have pocket money and they’re hiding and then they buy it, but otherwise they don’t have the funds they don’t have unless they go walking. But really, it’s, I think the parent and if you’ve made healthy food at home or some other alternatives, there’s no need for them to buy”—P10, (F), 43 y, >1 adolescent “…, the only soft drink I do bring. … my kids actually prefer it anyway Is the sparkling water so I get sparkling mineral water. I get it a little bit flavoured. That’s the only soft drink they do drink and they only get soft drink on that rare occasion where there It’s like a family barbecue … but it’s not something I go to the shop and say okay I need to buy coke every day I need to buy it on a weekly basis for that reason. Whereas for example, if I saw the mineral water on special I’ll buy by a box, because I know I like I said it’s the better alternative for them”—P21, (F), 39 y, >1 adolescent
**Psychological Capability: Skills**
“Look, I don’t mind fast food but it’s got to be like I said, either Subway, Lewrap or like Douggies grill we like their salad We I mean, whenever we go there we only get the chicken salad and sweet potato. Like if I can’t cook for some reason, I would go for that kind of food. Like salad type, like salad, like things that are healthy and, you know, it’s healthy”—P3, (F), 39 y, 1 adolescent “Yeah I’ll make burgers at home. I’ll make like if they say they feel like burgers I’ll make it for them instead. And then I add the extra tomatoes and the lettuce and the stuff like that to it”—P12, (F), 48 y, >1 adolescent “but I know we are a lot busier than we were and takeaway is cheaper sometimes and getting all the ingredients. So I think as time management and by the time you’ve bought all the ingredients, I could have just bought a takeaway meal and then everyone will just have the meal and there’s no dishes and it’s done and…even if I’m making spaghetti, like, I’ll grate, it really finely. And they’ve got no idea it’s veggies in there, but if I get bolognaise sauce that’s already pre-made I’ll make sure to get one that has veggies So if I make a meal that doesn’t have a salad on the side…. Yeah, I can get myself organized and meal prep and all that. Then we probably would never have to. It will be like a treat. By your time factor here…”—P17, (F), 43 y, >1 adolescent “So I prefer to just do home cooked meals or we go to Thai food, I try to encourage, you know, the cuisines that have like more vegetables”—P19, (F), 45 y, 1 adolescent
**Physical Capability: Skills**
“Myself Yes. When I’m sick, I just can’t bear to get up or do anything. So sometimes it’s just having a little bit of reliance on the two older kids to do that for the younger ones, I do feel guilty that I’m unable to do so. But as you said, there are circumstances sometimes that arise that you’re unable to do that for them and things like that. So there is a little bit of guilt going through that. I could have done it. But yeah, I’ve got two other older ones that are able to help me out in that regards”—P4, (F), 32 y, >1 adolescent
**Reflective Motivation: Beliefs about Consequences**
“They would be really overweight, especially you know fast food, soft drinks. I think they’re the two biggest culprits in making kids fat”—P19, (F), 45 y, 1 adolescent “As in not like not having breakfast oh yeah because you get the stomach grumbles, you get the dizzy spells, you can’t focus, you can’t concentrate, you know, you get, like, agitated because, you know, you’re hungry like your body tells you”—P14, (F), 43 y, 1 adolescent “Yeah. So I think obviously if they’re not having the right nutrition, then they’re putting on the weight because they’re going to eat all processed food… And obviously they’re gaining weight, because if you’re not having, you know, say five, four pieces of fruit, or sorry 5 pieces of veggies, seven, whatever it is that you’ve got to have intake, and then not having the proper fruit intake, then, you know, there’s not much nutrition going into the body, which sort of leads to other things like constipation or whatever”—P9, (F), 42 y, 1 adolescent “I think it would have a very negative effect on their health. I think weight gain is probably the first one. And just I think I don’t know if you really call it an addiction, but I think sugar is addictive. I think if it’s there they’re they’re going to they want they’re going to want it. Definitely. If we go to the supermarket, you know, my kids will still ask for lollies. They’ll still say, oh, can we get a bag of lollies or can we get these sweets or can we? And I’ve got to always kind of, you know, sometimes I’ll allow it, it’s most of the time I put my foot down and say no choose something alternative to that. I’m the boring mom, o you got to have fruit”—P7, (F), 42 y, 1 adolescent
**Reflective Motivation: Social/Professional Role and Identity**
“regarding their health. Yep health is really, really important in our family. So I try and set an example with my children and the things that we eat in the way that I shop sometimes, the way that I even speak about food. And it’s really important for me as a mother to make sure that I set up a healthy, healthy relationship with food as well, because I’m cognizant of the fact that there are a lot of eating disorders... Frankly, I’m aware of the fact that with teenagers in general, this is something that even I witnessed as a teenager myself, that there’s an unhealthy relationship with food that comes from different sources. I think whether it’s, you know, the idealistic view of women and the way they’re portrayed in media and in fashion... Sometimes, you know, the parent or the way that, you know, attitudes towards food play out home. So I’ve always been really, really aware of that. And I just I try and make sure that I’m setting a good example for the children and setting them up for adulthood to enable them to make the right decisions when it comes to their diet and their lifestyle. Inshallah.”—P7, (F), 42 y, 1 adolescent “Yeah, I definitely think that the government has a huge role, obviously, by subsidizing, you know, fruits and vegetables and things like that, you know, putting taxes on unhealthy food. I think that’s a big part, obviously, at school as well, like some of the school canteens, are shocking in what they sell, you know, and then obviously in educating as well, the students and having programs, especially for them. Yeah. As community groups, I don’t know how they can do much. I mean, my kids have gone to the go4fun programs that are run, and they did benefit for it, you know, from for a little while. But I feel like, you know, it just like with anything, they go through phases. So they’ll be all into it for a little bit and then it just sort of dies out.”—P18, (F), 40 y, 1 adolescent “… educate them when they’re younger, like still in like before they even start high school when they first start primary school. That’s when they should be educated, but not when they’re older and it’s too late. They’ve got into a routine and in a habit with their food and stuff”—P12, (F), 48 y, >1 adolescent
**Reflective Motivation: Goals**
“for it to be my role. Now at their age I’d probably say a nine. If they were older and they knew and you know more aware and more responsibility and things like that obviously it would be less important for me to take on that role and more for them. But at this point, at this age, oh I’d probably say nine that it’s my responsibility”—P5, (F), 37 y, 1 adolescent
**Reflective Motivation: Optimism**
“I think they totally do know the importance because they do it in school also in PDHPE and all that kind of stuff, but whether they follow through on it is another thing. I mean, so they know totally, but it’s just getting them to do it”—P19, (F), 45 y, 1 adolescent
**Automatic Motivation: Emotion**
“I worry with the kids more than us. I don’t want them to get into the habit that we’re in where they’ll skip meals, because once you skip breakfast, you tend to skip lunch and then you’ll be like, I’ll just have a big dinner and then snacks throughout the night, which is not good.”—P14, (F), 43 y, 1 adolescent “And once the child gets a taste of it, basically it just becomes a obsession because it keeps them on a high because of the sugar intake and the sweetness of them”—P4, (F), 38 y, >1 adolescent “One hundred percent. I worry about his eating because he does not, he doesn’t make good choices. He will always go for the unhealthy option. He does not like vegetables. The only vegetable he will eat is a carrot or a cucumber and carrots that’s pushing it cucumbers every now and again.... Fruit, he loves all fruits of any fruit you put in front of him, he will eat, which is at least he likes fruit. And when it comes to making good choices with the food that he eats, no, I am quite concerned for his health. And if he was not a tall boy, I think he would fill out quite quickly. But I think his height helps him a lot in his appearance”—P16, (F), 38 y, 1 adolescent
**Automatic Motivation: Optimism**
“Yeah, they know, because we talk a lot about, you know, the diet and we talk a lot about the importance of food and to keep yourself healthy. And also we do mention like eating. Now, when you are a teenager, it’s a reserve for when you get older”—P22, (F), 50 y, >1 adolescent
**Automatic Motivation: Reinforcement**
“No. Definitely no reward or punishment. I have tried to stay away from reward and punishment with regards to food, because I think that sets up an unhealthy relationship with the food, though, but I just I just enforce it. It’s basically just mums would have to have breakfast, you’re not, leaving the house until you have something. So yeah”—P7, (F), 42 y, 1 adolescent “I just don’t buy it if they wanted to have it I don’t buy it or I just hide it. I had to actually hide the whole bottle, if I think they’ve had too much, I just I like I usually get more on Sundays if I think they’ve had it too excessively, my punishment, but without them realizing what I’d take away the whole soft drink and put it somewhere else completely where they can’t find”—P10, (F), 43 y, >1 adolescent
**Social Opportunity: Social Influences**
“I probably have to agree, to be honest, because growing up here in Australia, I was also born and raised here, but I’m from the Middle Eastern background. Everything revolved around food, everything. Our parents, it was all about food you had eating was equal to being happy. I think that’s what our parents instilled in us. And I when I started to have my children started to see it that way. But then as they got older and I got wiser and I thought, no, but it’s you know, food does not equal happiness. It also equals overweight and obesity and the health problems and all these other things that come along with it. So I have to I tend to agree with those statistics, I think”—P16, (F), 38 y, 1 adolescent “And I think a big part of it also comes from culturally, you know, you can’t starve your kid, you know what I mean? You don’t want him to be upset or, you know, or like it’s slack to let them be hungry and that sort of thing. So it doesn’t surprise me, but I didn’t think it would be that big a difference”—P18, (F), 40 y, 1 adolescent “Yes, 100 percent. He always wants to eat the stuff that his friends like, they’re always sending each other snapshots on the latest place that’s opened up and what’s on the menu. And they should go try this and should go try that and he harasses me relentlessly he wants me to drive to woop woop to get a burger or a kebab. I’m like, I’m not driving like forty-five minutes from home, So you can try a new burger flavor or new chips dipped in you know, caramel or whatever it is like I’m not doing that. But yeah, they do, they play a massive role, she’s the same. She wants to go to all these weird places that her friends, have told her about where they can try all these new sweets and desserts. And I’m like, you know, it’s not happening”—P14, (F), 43 y, 1 adolescent “I guess at the moment I’m having a conflict with my fourteen-year-old son. He wants to try having a V. Even he doesn’t really drink soft drinks so that because the influence of his friends they are drinking that drink that he should be able to drink that drink too. And like it’s no its not working that way”—P4, (F), 38 y, > 1 adolescent
**Physical Opportunity: Environmental Context and Resources**
“No, no, and neither they’ve never outreached to us, have not in high schools, Oh they had Oh no Crunch and Sip was in primary school. They don’t do that in high school. I was going to say that they do a program called Crunch and Sip in primary school, which is where they take a piece of fruit and they physically have fifteen-minute break. But they don’t do that in high school”—P14, (F), 43 y, 1 adolescent “I think they should crack down on Uber eats at school time for school kids… I think teachers need to keep more of an eye on things like that. … You can get schools around shops, and they’ve got all those junkie shops like as in Kebabs and Pizza and KFC and Maccas. But when they’re far away from those kind of things, they’re not going to be tempted to go eat it from there. These days they can do Uber eats”—P3, (F), 39 y, 1 adolescent “Well schools really tick me off because I never knew that, Like, I know a lot of schools would ban it but then I found out that the high school actually sold soft drinks to the students so I mean, like, that’s so wrong. So, yeah, schools should definitely play a part. It’s very important…You can regulate as much as you can. But like I told you, what they choose to do. Like my daughter, if she chose to walk to Maccas it’s right next to her school …. So do you get what I mean, like, it’s always there next to you. The temptation is there, I guess for them, like I would say, it’s harder to regulate”—P19, (F), 45 y, 1 adolescent “…The kids lose trust in their parents because they see that the only mother is acting this way and the whole environment are the opposite. So that’s why I say it’s a big struggle…Yes, sometimes tight with the money, because the food is not cheap. The vegetables is not cheap you cannot provide every single day everything now…, I think time for working Mother even for working, Mother. It’s just it’s too much commitment, but it’s more think financially you feel that you cannot provide every single day.”—P22, (F), 50 y, > 1 adolescent

## Data Availability

Owing to privacy reasons in ethics, we cannot make the data publicly available, but please contact the author if further information is required.

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
