# Peer review of "Enabling Better Nutrition for Adolescents from Middle Eastern Backgrounds: Semi-Structured Interviews with Parents"

_nutrients, 2021, doi:10.3390/nu13113918_

Round 1
Reviewer 1 Report
Thank you for giving me the opportunity to review: Enabling better nutrition for adolescents from Middle Eastern backgrounds: semi-structured interviews with parents. This is a very well written paper with well-described methods and results. The findings will be of great interest to those working in the area of childhood/adolescent weight management and interventions, especially with families from minority backgrounds.
I have just one suggestion to add to the paper and have highlighted some minor typos.
P.3, lines 127 – 130: I realise you are using the word ‘it’ to mean the interview but it reads a bit strange, are you able to substitute with ‘they’ as to mean the researcher?
- 8, Psychological Capability/knowledge section in the quotes the drink ‘V’ is mentioned, for those who do not know what this is can you include: (branded energy drink).
P13, line 202: a space is need between ‘SSB’ and ‘and’
Lines 203 – 204: change ‘soft drink’ to ‘soft drinks’
Finally, you briefly mention your other work, focus groups conducted with ME adolescents, I appreciate that the current paper is focusing on the parents but it would be interesting to compare some of the results from both papers, e.g if there is any commonalities? It is interesting that although parents speak about lifestyle interventions, there was very little conversation around encouraging the adolescents themselves to have a greater input to their diet, cooking in the home etc with the family (one point made was older siblings helping younger ones with breakfast), might be worth a little more exploration/discussion.
Author Response
Reviewer 1
Thank you for giving me the opportunity to review: Enabling better nutrition for adolescents from Middle Eastern backgrounds: semi-structured interviews with parents. This is a very well written paper with well-described methods and results. The findings will be of great interest to those working in the area of childhood/adolescent weight management and interventions, especially with families from minority backgrounds.
Thank you very much for this comment. We are glad that you consider this work will contribute to serving at-risk communities.
I have just one suggestion to add to the paper and have highlighted some minor typos.
P.3, lines 127 – 130: I realise you are using the word ‘it’ to mean the interview but it reads a bit strange, are you able to substitute with ‘they’ as to mean the researcher?
Thank you, this has been amended and ‘it’ has been substituted to ‘they’.
- 8, Psychological Capability/knowledge section in the quotes the drink ‘V’ is mentioned, for those who do not know what this is can you include: (branded energy drink).
Thank you, this has been amended and have included ‘branded energy drink” in the quote.
P13, line 202: a space is need between ‘SSB’ and ‘and’
Thank you, this has been amended.
Lines 203 – 204: change ‘soft drink’ to ‘soft drinks’
Thank you, this has been amended.
Finally, you briefly mention your other work, focus groups conducted with ME adolescents, I appreciate that the current paper is focusing on the parents but it would be interesting to compare some of the results from both papers, e.g if there is any commonalities? It is interesting that although parents speak about lifestyle interventions, there was very little conversation around encouraging the adolescents themselves to have a greater input to their diet, cooking in the home etc with the family (one point made was older siblings helping younger ones with breakfast), might be worth a little more exploration/discussion.
Thank you for this suggestion, we have included an additional paragraph in the discussion outlining such similarities. Please see lines 517-530.
Reviewer 2 Report
Dear Authors,
Please find below comments and suggestion usefull to make themanuscript stronger:
- Abstract: For a reader, it is helpful to identify more precisely where the survey was performed.
- Introduction: In this section, you should specify the age range of adolescents the survey refers to. In older adolescents, the influence and role of parents is more limited.
- In this section, we would like more information on what basis the distinguishing features of proper nutrition (eating breakfast, vegetables and fruits, drinks and fast food) were selected. For example, was the consumption of sweets (whole-grain products, dairy) omitted?
- Can the acquired group of respondents be a representative group?
- Was the used interview questionnaire previously validated?
- I think the term "soft drinks" should be more clarified. Probably it was about sweetened / sugar drinks?
- What type of fast-food dishes are typically eaten / popular in the assessed population.
Author Response
Dear Reviewer,
Thank you for your comments. Please see below a point-by-point response to address your concerns.
Reviewer 2
Dear Authors,
Please find below comments and suggestion usefull to make the manuscript stronger:
- Abstract: For a reader, it is helpful to identify more precisely where the survey was performed. This has been amended to include “in Australia” please see line 10.
- Introduction: In this section, you should specify the age range of adolescents the survey refers to. In older adolescents, the influence and role of parents is more limited. Thank you this has been amended to include the age range of13-18 adolescents, please see line 67.
- In this section, we would like more information on what basis the distinguishing features of proper nutrition (eating breakfast, vegetables and fruits, drinks and fast food) were selected. For example, was the consumption of sweets (whole-grain products, dairy) omitted? Thank you for this comment. The study topics for the interview questions were based on the key nutrition behaviours that ME adolescents were more at risk of not reaching, as identified by the SPANs survey as identified in the introduction. We have included a sentence clarifying this under 2.3. Procedure and Data Collection. Please see lines 132-133.
- Can the acquired group of respondents be a representative group? Thank you for this comment. As with all qualitative research the results are generalizable for this group but not beyond that. The group is typical of Middle Eastern families – residing in the mostly Sydney suburbs they tend to concentrate (that are low SES status), and with two parent families.
- Was the used interview questionnaire previously validated? The interview questionnaire was designed by two qualified researchers with one from Middle Eastern background. The questionnaire was previously tested for interview duration and understanding of the questions. The questions are based on the well-known COM-B and Theoretical Domains Framework widely used in formative research for design of behaviour change programs.
- I think the term "soft drinks" should be more clarified. Probably it was about sweetened /sugar drinks? This has been expanded to carbonated sweetened beverages. Please see line 132. The interviews generated discussion around other drinks such as energy drinks (V) and fruit beverages which we included under “Sugar Sweetened Beverages’ in the results.
- What type of fast-food dishes are typically eaten / popular in the assessed population. Thank you for this comment. Popular fast food includes McDonalds, Oporto’s (which is a chicken sandwich/burger outlet) and KFC (fried chicken). We also included fast food deemed alternatives such as charcoal chicken and Lebanese restaurants to be popular in this population. This information was included in the results line 220.